# Connectomics of the zebrafish's lateral-line neuromast reveals wiring and miswiring in a simple microcircuit

Eliot Dow[1,2], Adrian Jacobo[1], Sajjad Hossain[1], Kimberly Siletti[1], A J Hudspeth[1]*

[1]Laboratory of Sensory Neuroscience, Howard Hughes Medical Institute, The Rockefeller University, New York, United States; [2]Presence Saint Francis Medical Center, Evanston, United States

**Abstract** The lateral-line neuromast of the zebrafish displays a restricted, consistent pattern of innervation that facilitates the comparison of microcircuits across individuals, developmental stages, and genotypes. We used serial blockface scanning electron microscopy to determine from multiple specimens the neuromast connectome, a comprehensive set of connections between hair cells and afferent and efferent nerve fibers. This analysis delineated a complex but consistent wiring pattern with three striking characteristics: each nerve terminal is highly specific in receiving innervation from hair cells of a single directional sensitivity; the innervation is redundant; and the terminals manifest a hierarchy of dominance. Mutation of the canonical planar-cell-polarity gene *vangl2*, which decouples the asymmetric phenotypes of sibling hair-cell pairs, results in randomly positioned, randomly oriented sibling cells that nonetheless retain specific wiring. Because larvae that overexpress Notch exhibit uniformly oriented, uniformly innervating hair-cell siblings, wiring specificity is mediated by the Notch signaling pathway.

DOI: https://doi.org/10.7554/eLife.33988.001

*For correspondence: hudspaj@rockefeller.edu

**Competing interests:** The authors declare that no competing interests exist.

## Introduction

The assembly of the $10^{15}$ synapses of the human brain is a formidable challenge that begins in the developing embryo. Although the guidance of an axon to a general region of the nervous system restricts the search for an appropriate synaptic partner, there remain numerous cellular targets with which the axonal terminal could create a synapse. Environmental stimuli play a role in nervous-system development, but many neuronal connections nevertheless form normally in the absence of activity, suggesting the presence of additional mechanisms guiding the targeting of synapses. The zebrafish's lateral line represents a simple but powerful model system for exploring the mechanisms that guide the assembly of microcircuits.

Most fishes and many aquatic amphibians possess mechanoreceptive systems known as lateral lines, whose responsiveness to water currents underlies rheotaxis, predator avoidance, and schooling. Each lateral line consists of one to several neuromasts, which are discrete epithelial organs that include hair cells as well as the underlying supporting cells and surrounding mantle cells. Although lateral lines may be distributed over an animal's head, body, and tail in a variety of patterns, a general feature is directional sensitivity. Each hair cell possesses on its apical surface a mechanically sensitive hair bundle that is usually responsive to water movement either in the rostrad direction—toward the animal's front end—or in the caudad direction—toward the rear end. Vectorial sensitivity results from morphological polarization of each hair bundle, which consists of an array of actin-filled stereocilia whose lengths increase progressively in the direction to which the bundle is responsive.

In the larval zebrafish, directional responsiveness arises from differential innervation by two oppositely polarized subpopulations of hair cells, each attuned to the detection of water currents toward

either the animal's front end or its rear end. During early development, the primary lateral line on each side of a larva comprises seven neuromasts, discrete organs each containing 8–20 hair cells that make synapses onto afferent axonal terminals and receive synaptic inputs from efferent terminals (*Figure 1A and B*). In wild-type zebrafish there are approximately equivalent numbers of hair cells with hair bundles of the two polarities (*Figure 1C and D*). Throughout the life of a fish, hair cells of the two types arise continually as transit-amplifying cells divide to produce sibling hair cells of opposing polarity (*Figure 1E*).

The differentiation of new hair cells is mediated in part by lateral inhibition through the Notch signaling pathway (*Lanford et al., 1999*; *Wibowo et al., 2011*). Following mitosis, the newly formed sibling cells undergo a series of stereotypical rearrangements in their relative anteroposterior positions, after which the more anteriorly positioned hair cell develops a hair bundle sensitive to caudad stimulation and the posteriorly positioned sibling extends a bundle responsive to rostrad water movement. Because *trilobite* mutant larvae, in which the vangl2 protein is inactivated, possess hair bundles with random orientations, the canonical planar-cell-polarity pathway participates in cellular patterning. Finally, through a process that requires neither mechanotransduction nor synaptic activity, each afferent neuron of the posterior lateral line receives innervation from hair cells of only one orientation (*Figure 1F–H*; *Nagiel et al., 2008*; *Nagiel et al., 2009*; *Wibowo et al., 2011*). In addition to the linear arrangement of neuromasts along the tail, the neural calculations necessary for rheotaxis, escape swimming, and schooling require directionally specific information from discrete populations of neurons connected to hair cells of the two polarities (*Oteiza et al., 2017*).

Serial blockface scanning electron microscopy (SBFSEM) has made possible the reconstruction of complete axons and dendrites within modules of the nervous system (*Figure 1I and J*). Neuronal connections have been investigated in detail in species with very few neurons, such as roundworms, and in structures with a crystalline degree of order, such as the fruit fly's visual pathway. Although SBFSEM offers sufficient resolution to reveal the fine details of neural microcircuits in vertebrate nervous systems as well, the volumes of most structures of interest require weeks to months of specialized microscopy and tens of thousands of hours of manual segmentation to reconstruct a single specimen (*Denk and Horstmann, 2004*; *Helmstaedter et al., 2011*; *Wanner et al., 2016*; *Hildebrand et al., 2017*). As a consequence, there are few if any SBFSEM studies in which multiple circuits have been compared between individuals or during development. To gain insight into the assembly of the neural connections that allow the detection of directional stimuli, we have used SBFSEM to conduct a comprehensive description of neuromast wiring and an investigation of microcircuit assembly in wild-type and mutant larvae.

## Results

### Structure and innervation of wild-type neuromasts

To establish a basis for comparison with mutant specimens, we first sought to determine the complete connectome of the wild-type zebrafish's neuromast. We focused on the posterior lateral-line organs of larvae two to four days post-fertilization (2–4 dpf), a period during which the animals demonstrate behaviors dependent on a functional lateral line such as rheotaxis, escape swimming, and upright orientation. We analyzed single neuromasts from each of eight zebrafish (*Figure 2A*; *Video 1*). The axonal terminals associated with each neuromast entered through a single perforation in the basal lamina as branches arising from peripheral axons in the posterior lateral-line nerve (*Figure 2B*). Each SBFSEM data set included approximately 40 µm of a posterior lateral-line nerve. Because we did not collect long-range data relating each neuronal soma to its terminals, we wondered whether the terminals that contacted a specific neuromast could include two or more branches of the same axon. Upon examining 14 individually labeled neurons by confocal fluorescence microscopy, however, we found no instances of a bifurcated axon extending terminals into a neuromast. Whereas previous investigators had estimated that two to four neurons enter a neuromast (*Liao, 2010*), we found an average of 9.3 neuronal branches in each sensory organ.

The majority of neuromasts included one axonal terminal that made direct contact with every hair cell without apposition to any synaptic ribbon. Together with the presence of numerous presynaptic vesicles and postsynaptic membrane cisterns, this wiring pattern identified these structures as the terminals of efferent axons (*Figure 2C*). The remaining axonal terminals were considered to

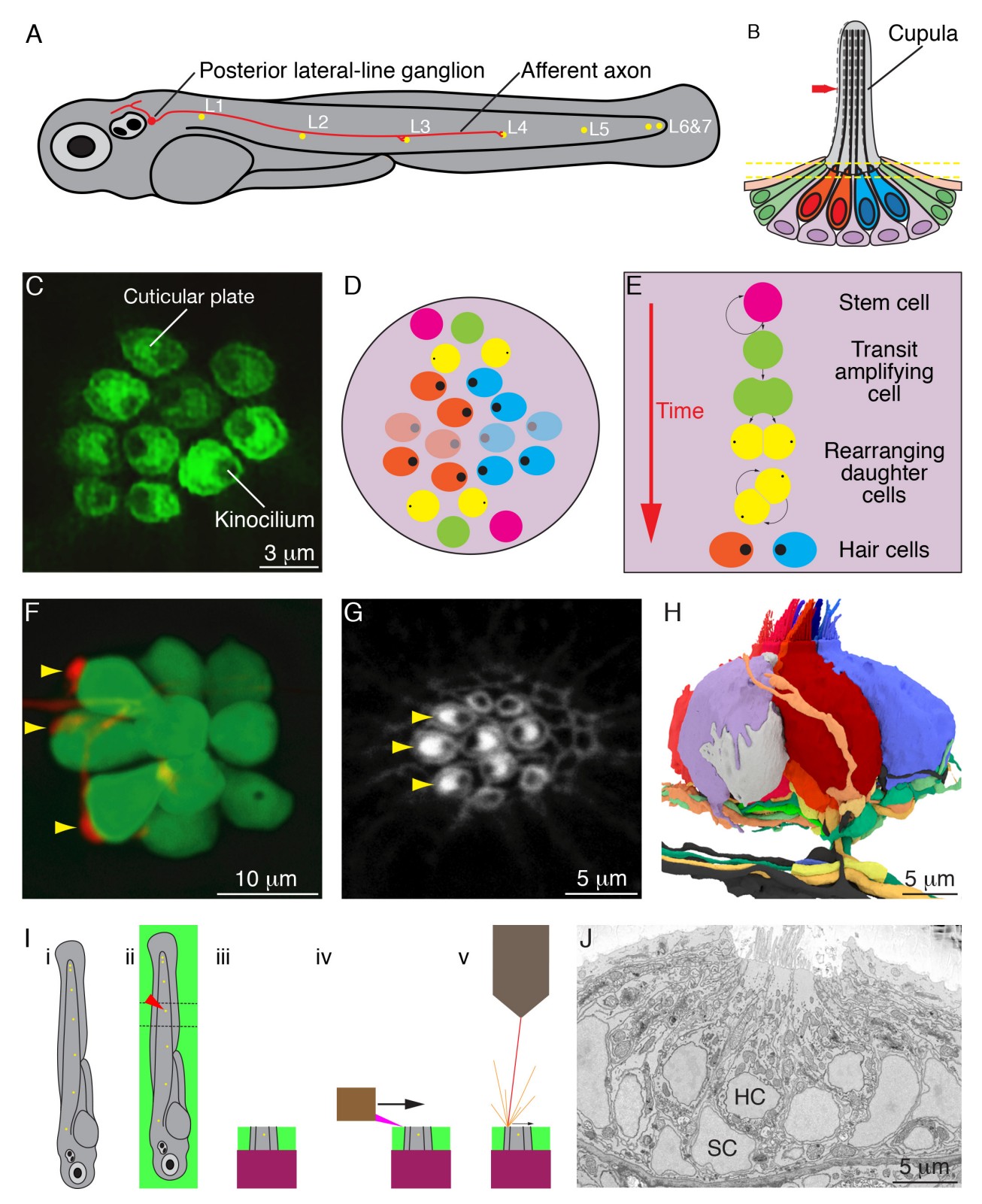

**Figure 1.** Organization of neuromasts and their innervation. (**A**) A schematic diagram depicts the posterior lateral line of the larval zebrafish. While migrating from the ear to the tip of the tail, the first lateral-line primordium deposits about seven neuromasts (yellow, *L1* through *L6 and 7*). The hair cells of these neuromasts are uniformly polarized along the anteroposterior axis; a second primordium subsequently deposits a few neuromasts (not shown) with a dorsoventral polarity. Individual neurons of the posterior lateral-line ganglion, one of which is portrayed, are innervated by hair cells in

*Figure 1 continued on next page*

*Figure 1 continued*

one to five successive neuromasts and convey information into the brainstem. (B) A schematic diagram of an individual neuromast shows a cluster of hair cells surmounted by a gelatinous cupula into which their hair bundles insert. When water movement (arrow) deflects the cupula, the long kinocilia are bent and their motion is communicated to and transduced by the stereocilia. Half of the hair cells (red) are polarized for sensitivity to caudad movement, the other half (blue) to rostrad motion. The hair cells are separated by supporting cells (lavender) and bounded by mantle cells (green). (C) A high-resolution, deconvolved fluorescence image shows the apical surfaces of hair cells in the plane of section indicated by parallel dashed lines in panel (B). The cuticular plate at the apex of each hair-cell soma is marked by actin-GFP. The dark spot within each cuticular plate represents the base of the kinocilium, which denotes the hair bundle's polarity. (D) A schematic diagram depicts the organization of hair cells in a neuromast of the posterior lateral line. There are roughly equal numbers of hair cells sensitive to caudad water motion (red) and to rostrad movement (blue). As hair cells migrate toward the neuromast's equator (pale cells), they senesce and die. (E) Uncharacterized stem cells located near the apical and basal poles of the neuromast produce transit-amplifying cells, each of which divides into daughter cells that undergo a rotatory rearrangement in about half the instances. After rearrangement has concluded, the nascent hair cells in wild-type larvae always adopt opposite polarities. (F) An individual afferent nerve fiber marked with mCherry forms postsynaptic endings (arrowheads) on three hair cells. (G) An image of the apical surface of the same neuromast shows phalloidin-labeled cuticular plates and hair bundles. Consistent with polarity-specific innervation, the three hair cells contacted by the fiber in panel (F) all display polarization to caudad stimuli. (H) A reconstruction of the innervation of a single neuromast includes afferent terminals, which receive synaptic input from sensory hair cells, and an efferent terminal that innervates hair cells. Details are provided in the caption of *Video 2*. (I) Steps in serial blockface scanning electron microscopy (SBFSEM) of a lateral-line neuromast. (i). A larva is preserved in aldehyde-based fixative and impregnated with heavy metal to enhance the specimen's contrast and electrical conductivity. (ii) The specimen is embedded in plastic and cut (dotted lines) to isolate the region containing the neuromast of interest (arrowhead). (iii) The trimmed specimen is secured to a mount and placed in the scanning electron microscope. (iv) A diamond knife makes hundreds to thousands of passes across the blockface, progressively scraping away tens of nanometers of the specimen. (v) After each passage of the knife, an electron beam scans the blockface; the emitted secondary electrons are collected to form an image of the sectioned specimen. (J) A typical serial blockface scanning electron micrograph shows several hair cells, whose nuclei (HC) occur in a layer above that encompassing supporting-cell nuclei (SC).

DOI: https://doi.org/10.7554/eLife.33988.002

represent afferent neurons. Of these terminals, 16% displayed minimal or no branching and tapered to termini immediately inside the neuromasts. The remaining 84% of the terminals occupied perisynaptic compartments, the spaces bounded by supporting cells beneath hair-cell synapses (*Figure 2D*); 74% displayed membrane contacts at hair-cell bases, and 59% were postsynaptic to hair cells at ribbon synapses.

## Specificity of innervation

In view of the relatively restricted number of afferent and efferent terminals, hair cells, and their associations, we represented the data from each neuromast as a schematic diagram showing ribbon synapses, membrane contacts, and perisynaptic associations between hair cells and axonal terminals (*Figure 3*; *Figure 3—figure supplement 1*; *Figure 3—source data 1*; *Video 2*). Expanding upon previous reports, our data revealed the striking polarity specificity of lateral-line neurons (*Nagiel et al., 2008*; *Dow et al., 2015*). Among 344 presynaptic active zones in hair cells, only four were apposed to axonal terminals of the opposite polarity preference. Furthermore, 92% of the contact area between hair cells and terminals was polarity-specific. This specificity was not predominantly a result of spatial segregation, for many axonal terminals were located perisynaptically to hair cells of both polarities. Moreover, specificity did not appear to arise simply by the precocious arrival of an appropriate afferent terminal at a nascent hair cell: in all six observed cases of young hair cells that contacted axonal terminals but had not yet developed ribbon synapses, the hair cells made predominant or exclusive contact with terminals whose innervation by neighboring mature hair cells indicated sensitivity to rostrad stimulation.

To assess the degree of neuronal wiring specificity throughout development, we used preimaging of neuromasts by fluorescence confocal microscopy to divide postmitotic hair cells into three temporal classes associated with characteristic cellular activities: 0–5 hr, when sibling pairs of hair cells undergo rearrangement along the anteroposterior axis; 5–15 hr, when the hair cells extend dynamic basal projections to mediate innervation; and more than 15 hr, when the formation of ribbon synapses largely concludes and the hair cells no longer rearrange or extend basal projections (*Wibowo et al., 2011*; *Mirkovic et al., 2012*; *Dow et al., 2015*). For those hair cells lacking preimaging data we used a machine-learning approach for classification into these three age categories, as described in the Methods. The formation of polarity-specific ribbon synapses between hair cells and afferent terminals began as soon as synaptic ribbons became apparent, with no significant

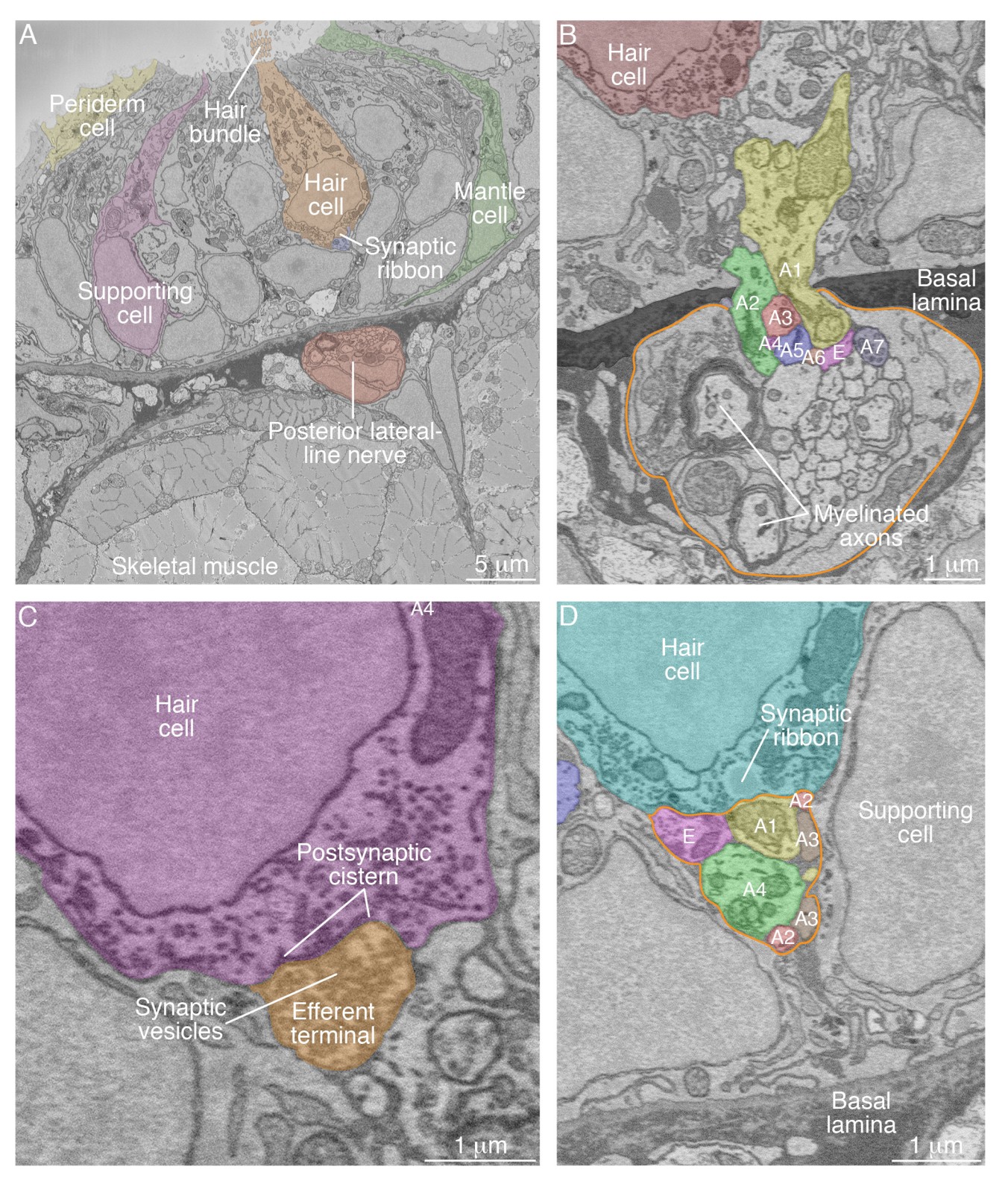

**Figure 2.** SBFSEM images of neuromast components. (**A**) A low-power micrograph shows the general organization of a neuromast. The sensory organ lies between skeletal-muscle fibers, from which it is separated by the epithelial basal lamina, and the aqueous environment, into which the hair bundles protrude. The nuclei of hair cells form a layer above those of supporting cells. (**B**) The posterior lateral-line nerve (outlined in orange) contains two myelinated axons as well as numerous unmyelinated fibers. Seven afferent terminals, each denoted as '*A*' and distinctively colored and numbered, arise

*Figure 2 continued on next page*

*Figure 2 continued*
as single branches of axons in the nerve and enter the neuromast through a pore in the epithelial basal lamina. One efferent axon labeled 'E' is also present. (C) The synaptic terminal of an efferent axon onto a hair cell displays a vesicle-filled bouton. Within the hair cell's cytoplasm lies a cistern that sequesters the $Ca^{2+}$ that enters the cell during synaptic activity. (D) A perisynaptic compartment (outlined in orange) contains four afferent terminals, each denoted as 'A' and numbered, and a single efferent terminal labeled 'E.' The compartment is demarcated by two supporting cells. The hair cell makes an afferent synapse onto the axonal terminal A1. A synaptic ribbon in the hair cell is characterized by moderate electron density and a clear halo surrounded by synaptic vesicles. The ribbon is readily distinguished from the numerous mitochondria in all cell types. In each of the panels, the areas within some of the cellular contours delineated by annotators have been colored to emphasize specific cells or axonal terminals. In order to provide a broad color gamut, the scheme of coloration here is arbitrary and distinct from that in the other figures.
DOI: https://doi.org/10.7554/eLife.33988.003

change over time in the frequency with which hair cells of the opposite polarity innervated the terminals. In addition, the preponderance of the contact area was occupied by polarity-appropriate axonal terminals by 5–15 hr, a bias that grew after 15 hr (*Figure 4A*).

By viewing the complete set of cellular interactions at ribbon synapses during different stages of synaptic development, we recognized that specificity is mediated primarily by direct interactions between hair cells and axonal terminals. A perisynaptic compartment contained as many as nine terminals, and the average number increased during maturation. Because no other cells intervened between the nerve fibers and the synaptic regions of hair cells, it is unlikely that a guidepost cell or other third party mediated microcircuit assembly as has been reported in other systems (*Shen et al., 2004*). This pattern suggested that the polarity specificity of innervation depends upon short-range cues unique to hair cells and the polarity-appropriate subset of afferent neurons.

## Redundancy of innervation

Although the neuromast connectome displayed a high degree of specificity, it also manifested two levels of redundancy. First, the average number of synaptic partners per hair cell increased from one in cells aged 5–15 hr to two for cells older than 15 hr (*Figure 4B*). Second, in addition to its immediate innervation partners, each hair cell formed increasing numbers of perisynaptic relationships with other axonal terminals. Nearly equal proportions of afferent terminals of each polarity preference occupied a hair cell's perisynaptic compartment, but of the contacting terminals that did not form ribbon synapses a greater number were of the opposing polarity. These diverse associations persisted as hair cells matured (*Figure 4B*).

## Dominance of innervation

The redundant innervation by hair cells was unequally distributed: one axonal terminal occupied the majority of the hair cells' presynaptic active zones and contact area. This dominance by a single terminal increased with age: the dominant terminal received 34% of the ribbon

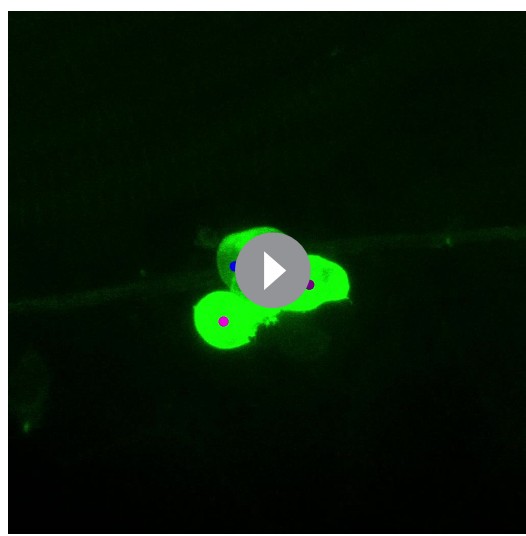

**Video 1.** Relation of preimaging by confocal fluorescence microscopy to SBFSEM images. The initial portion of the video depicts 31 hr of preimaging by confocal fluorescence microscopy of a wild-type neuromast (*WT4*). Each hair cell expresses green-fluorescent protein and is denoted by a uniquely colored dot. Note the extension of dynamic processes from the newly formed hair cells at the upper and lower poles of the neuromast, an activity associated with the formation of ribbon synapses. At the conclusion of preimaging, the specimen was placed in fixative solution prior to subsequent SBFSEM preparation and imaging. As shown at the conclusion of the video, there is clear correspondence between the preimaged hair cells and their representations in the resultant SBFSEM data. Dynamic processes are apparent on the bases of the white and especially of the gray hair cell. In order to provide a broad color gamut, the scheme of coloration here is arbitrary and distinct from that in the other figures.
DOI: https://doi.org/10.7554/eLife.33988.004

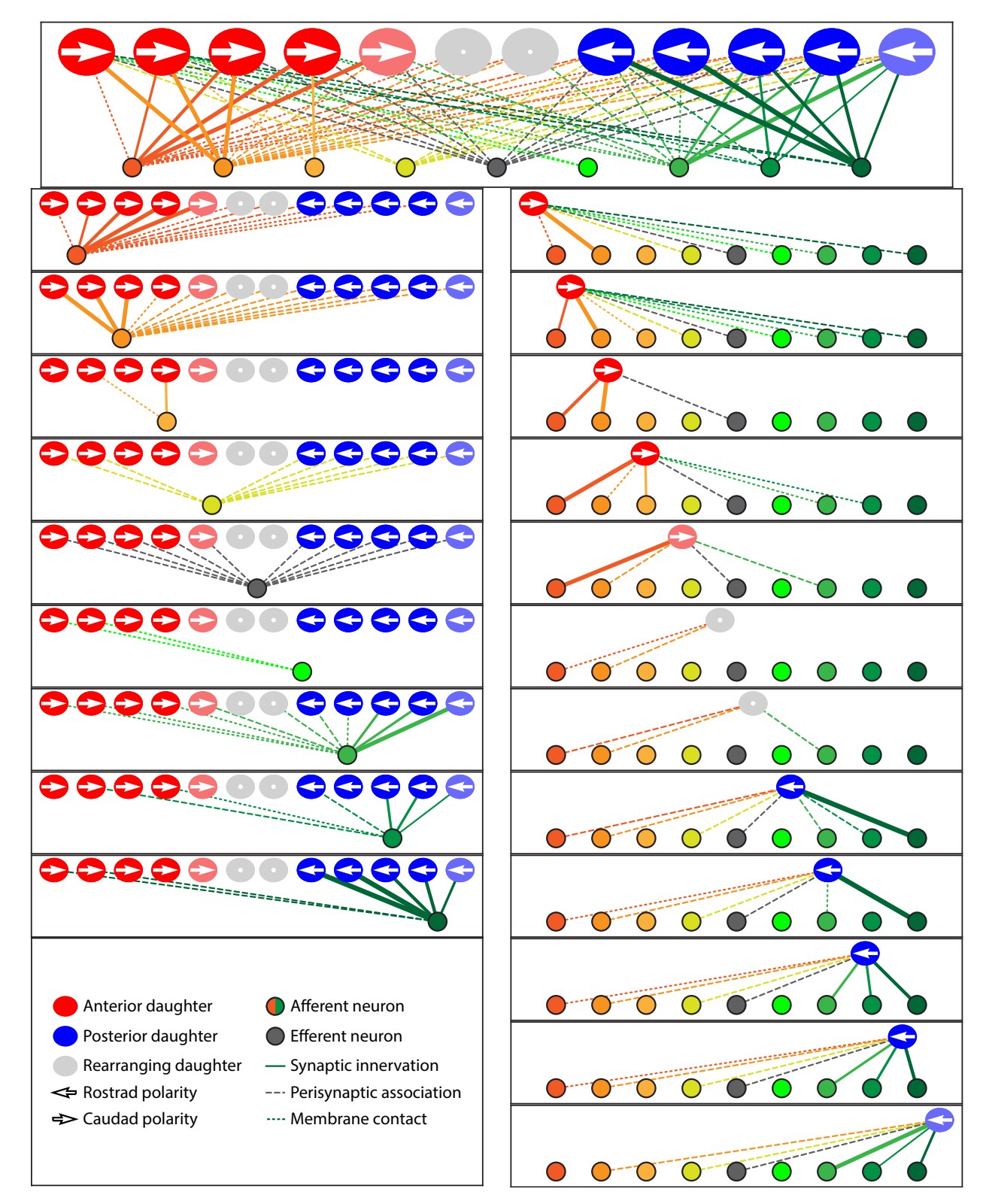

**Figure 3.** Microcircuit connectivity of the neuromast in wild-type zebrafish. The diagram displays the complete set of associations between hair cells and axonal terminals within an individual neuromast (*WT1*), including ribbon synapses (solid lines), membrane contacts (coarsely dotted lines), and perisynaptic associations (finely dotted lines). Hair cells are divided into two groups according to their anteroposterior positions following rearrangement: more anterior hair cells are red, more posterior hair cells are blue. The gray cells shown here and in supplemental diagrams represent

*Figure 3 continued on next page*

*Figure 3 continued*

hair cells 0–5 hr post-mitosis, which have not developed a definite polarity. White arrows define the hair-bundle orientations. Both subpopulations are further ordered from left to right by age, with those greater than 15 hr post-mitosis lying to the left of those 5–15 hr post-mitosis and bearing a darker hue of red or blue. Sibling cells occupy the same left-to-right positions within their respective populations. Axonal terminals are ordered from left to right based upon whether they form ribbon synapses predominantly with the more anteriorly positioned or the more posteriorly positioned hair-cell subpopulation. The terminals of a single efferent neuron are colored dark gray, whereas the other terminals represent afferents. The top panel depicts the full set of associations constituting the neuromast's microcircuit. The left column depicts separately the set of associations for each axonal terminal, whereas the right column depicts the associations of each hair cell. The lines representing terminals making ribbon synapses are graded in width in uniform steps representing from one to six synapses.

DOI: https://doi.org/10.7554/eLife.33988.005

The following source data and figure supplement are available for figure 3:

**Source data 1.** The tables provide a comprehensive set of measurements and calculations involving the data included in the main text and in *Figure 3—figure supplement 1*.
DOI: https://doi.org/10.7554/eLife.33988.007
**Figure supplement 1.** The diagrams depict the complete set of associations between hair cells and axonal terminals from seven additional wild-type neuromasts (*WT2-8*).
DOI: https://doi.org/10.7554/eLife.33988.006

synapses and 53% of the contact area of hair cells at 5–15 hr compared to 75% of the synapses and 73% of the contact area of hair cells older than 15 hr (*Figure 4C*). Furthermore, every neuromast possessed one terminal of each polarity that accounted for more than half of the presynaptic active zones and contact area for hair cells of that polarity (*Figure 4D*). This pattern suggests that dominance is a property of the axonal terminal that leads to its participation in the majority of ribbon synapses across a majority of hair cells. The average neuromast additionally harbored a second terminal of each polarity with roughly half the synaptic complement and contact area of the dominant one, and a third and sometimes fourth terminal that reached only a handful of presynaptic active zones.

Previous research has shown that lateral-line afferent neurons contact three to five consecutive neuromasts and often do so unequally, with some axonal branches ramifying extensively throughout a neuromast but other branches entering more distal or proximal neuromasts with minimal arborization (*Nagiel et al., 2008*). This pattern suggests that the dominance of axonal terminals is a property local to particular axonal branches and the corresponding neuromasts, rather than a characteristic of an entire neuron. Although the redundant, hierarchical neuronal associations may enable each hair cell to maintain innervation in the event of neuronal compromise, the presence in a neuromast of several terminals of graded sensitivity also offers the possibility of more refined stimulus detection by the sensory organ.

Considering that the formation of ribbon synapses appeared to arise from a combination of short-range, polarity-specific cues and interneuronal competition for dominance, we wondered how the two neuronal characteristics of dominance and specificity interact. When all fibers save one have been experimentally ablated, hair cells are able to innervate neurons of the opposing polarity (*Pujol-Martí et al., 2014*). Nevertheless, we encountered numerous instances in which the least dominant afferent terminal specific to a given polarity of hair cell participated in a ribbon synapse despite the contiguity of the most dominant terminal of the opposing polarity. This observation suggests the primacy of specificity over dominance. In support of this conclusion, there were no cases in which dominance superseded specificity in innervation, that is, in which the highest-dominance axonal terminal of the opposing polarity preference displaced the lowest-dominance terminal from a hair cell of its characteristic polarity preference. Within each polarity-specific subset of neurons, competition therefore determines the innervation partner.

Taken together, these data indicate that the microcircuitry of neuromasts is characterized by specificity, redundancy, and dominance. Polarity-specific neuronal associations begin at the earliest appearance of synaptic ribbons and increase with age along with the dominance hierarchy of axonal terminals and the redundancy of both innervated and contacted fibers. Each hair cell recruits a few polarity-specific neuronal partners while maintaining an extensive but loose network of diverse neuronal associations.

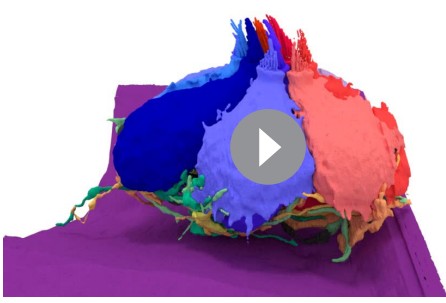

**Video 2.** Animated reconstruction of a wild-type neuromast. This neuromast (*WT1*) contains ten mature hair cells with hair bundles as well as a pair of nascent hair cells (lavender and gray) that have completed their rearrangement but not yet separated and formed hair bundles. Lateral views demonstrate the beveled hair bundles with stout kinocilia at their tall edges. Hair cells are colored according to their anteroposterior positions following rearrangement: more anterior hair cells in reddish shades and more posterior hair cells in blue. The terminals that form ribbon synapses with the former are delineated in various hues of yellow and orange and those associated with the latter are shaded green and blue. The single efferent terminal is black. All the axons extend their terminal processes though a single orifice in the basal lamina, which is colored purple. The spaces between the hair cells, and those around which the axonal terminals are arrayed, represent the positions of unseen supporting cells. In the second half of the sequence all the hair cells save one are removed to provide a view of the pattern of innervation, which is then further restricted to show the single dominant terminal.

DOI: https://doi.org/10.7554/eLife.33988.008

## Innervation of trilobite mutant neuromasts

Having delineated the characteristics of the neuromast's normal microcircuitry, we sought to determine how the arrangement arises by investigating zebrafish mutations that disrupt the polarity of hair cells. We first examined one neuromast in each of four *trilobite* mutant larvae, in which inactivation of the canonical planar-cell-polarity protein vangl2 causes hair bundles to orient randomly (*Hammerschmidt et al., 1996*; *López-Schier and Hudspeth, 2006*). In the absence of binary hair-bundle orientation, we used time-lapse confocal microscopy to score the specificity of innervation by determining whether a given hair cell completed cellular rearrangement anterior or posterior to its sibling, a behavior that in normal zebrafish corresponds perfectly with sensitivity to respectively caudad or rostrad stimulation (*Mirkovic et al., 2012*). By this criterion, axonal terminals of mutants did not show the normal pattern of discrimination: 73% of ribbons were partnered with terminals that were innervated both by hair cells that completed rearrangement anterior to their siblings and by those that assumed posterior positions (*Figure 5*; *Figure 5—figure supplement 1*; *Figure 5—source data 1*; *Video 3*).

If the cue that instructs specific innervation by hair cells were independent of the signaling pathways that mediate anteroposterior positioning and hair-bundle polarization, we reasoned that each sibling hair cell might nevertheless achieve the distinct chemical identity necessary to provide specific innervation. Indeed, when innervation was compared between sibling hair cells, we found that afferent terminals were almost perfectly innervated by only one hair cell per sibling pair; only one synaptic ribbon of 101 was apposed to an axonal terminal that also contacted the sibling hair cell. Because the more anterior or more posterior sibling did not preferentially innervate the terminals, the two chemical identities appeared to have been distributed randomly. This pattern indicates that, in the absence of canonical planar-cell-polarity signaling, the distinct cellular identity of each sibling that determines polarity-specific innervation is nevertheless uniquely allocated. However, unlike the hair cells of wild-type specimens, the siblings of a common cellular identity were randomly arranged with respect to their anterior-posterior position. Apart from specificity, the remainder of the *trilobite* microcircuitry remained intact: mutant specimens showed redundant innervation and a distribution of afferent-terminal dominance similar to that of normal larvae.

The *trilobite* specimens also allowed us to evaluate the role of electrical activity in synapse formation by hair cells. Previous research had shown that activity is not necessary for polarity-specific innervation, a rebuttal to the proposition that axonal terminals achieve specificity by detecting coincident depolarization of hair cells (*Nagiel et al., 2009*). However, it remained uncertain whether coincident synaptic activity could nonetheless suffice for specific innervation. The randomly oriented hair bundles of the *trilobite* specimens allowed us to test this hypothesis. We used preimaging with confocal fluorescence microscopy and subsequent SBFSEM to evaluate 14 neurons that were innervated by more than one hair cell. Each eschewed the formation of synapses with hair cells with a

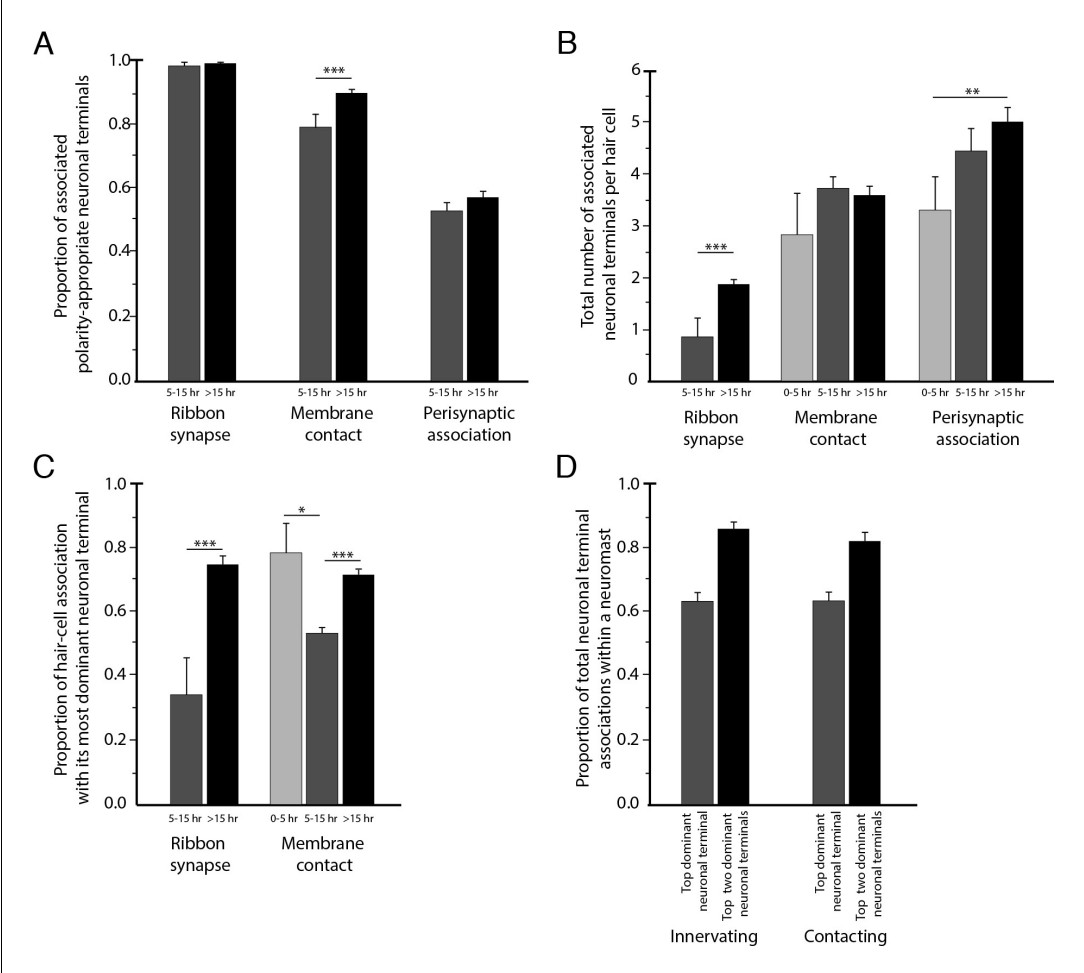

**Figure 4.** Statistical characteristics of the neuromast connectome. (**A**) A histogram depicts the proportion of ribbon synapses, membrane contacts, and perisynaptic associations formed with axonal terminals of the appropriate polarity for hair cells of 5–15 hr post-mitosis and for more than 15 hr post-mitosis. (**B**) A histogram depicts the total number of axonal terminals per hair cell that form ribbon synapses, membrane contacts, or perisynaptic association for hair cells of 0–5 hr, 5–15 hr, or more than 15 hr post-mitosis. (**C**) For each hair cell, the axonal terminal receiving the greatest number of ribbon synapses and making the largest area of contact with hair cells was designated as the dominant neuronal arbor. The histogram displays the proportion of a hair cell's ribbon synapses and membrane contacts associated with that terminal for hair cells of 0–5 hr, 5–15 hr, or more than 15 hr post-mitosis. (**D**) For each neuromast, neuronal arbors were ranked by the proportion of total ribbon synapses formed with the polarity-specific subpopulation. The histogram shows the proportion of ribbon synapses and afferent membrane contacts within a neuromast provided by the most dominant neuronal arbor and by the two less dominant arbors.
DOI: https://doi.org/10.7554/eLife.33988.009

similar vector of mechanosensitivity in favor of those with dissimilar or even opposite orientations. Synaptic activity is therefore neither necessary nor sufficient to instruct polarity-specific innervation.

## Innervation of Notch-overexpressing neuromasts

Finally, we analyzed polarity-specific innervation in a line of transgenic zebrafish that lack normal signaling through the Notch pathway. In the cochlea, Notch signaling mediates the asymmetric differentiation of hair cells and supporting cells through lateral inhibition as progenitor cells undergo mitosis (*Lanford et al., 1999*). Notch signaling is also active in the neuromast and appears to regulate the entry of supporting cells into mitosis to create new hair cells (*Haddon et al., 1998*; *Wibowo et al., 2011*). Previous work suggested that Notch signaling also plays a role in the polarization of hair cells, for disruption of the pathway with the γ-secretase inhibitor DAPT results in a biased distribution of hair-bundle polarizations (*Mirkovic et al., 2012*).

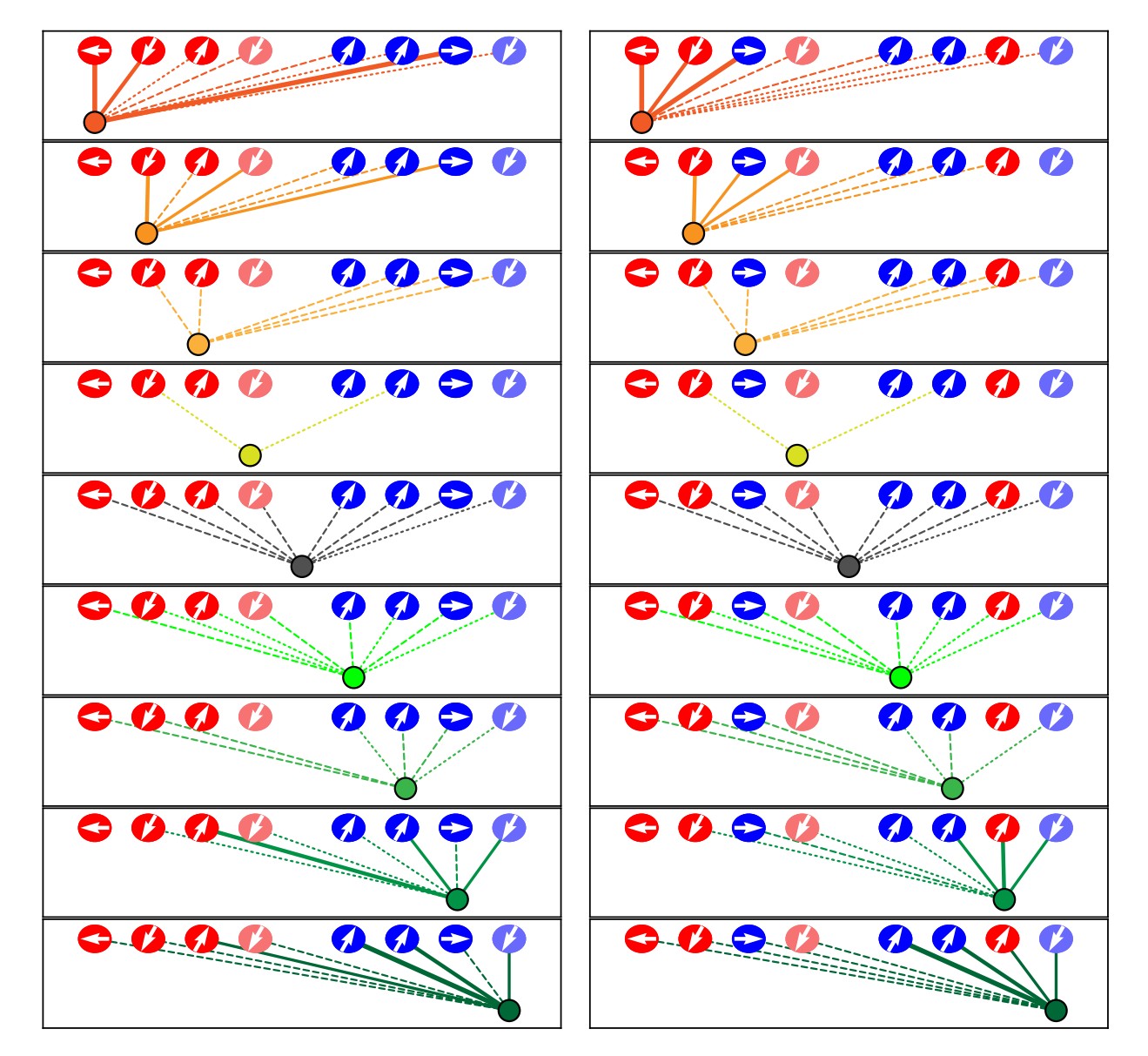

**Figure 5.** Microcircuit connectivity of a neuromast in a *trilobite* zebrafish. The diagram displays the complete set of associations between neuronal arbors and hair cells in a *trilobite* neuromast (T1). Note the erratic orientations of the hair bundles' axes of mechanosensitivity in this mutant of the planar-cell-polarity pathway. In the left column, the arrangement and symbols are identical to those of *Figure 3*. Hair cells are divided into two groups according to their anteroposterior positions following rearrangement: more anterior hair cells are red, more posterior hair cells are blue. In the right column, the same data are shown after repositioning of a single pair of hair cells known to be siblings to show that the innervation pattern still consists of two mutually exclusive set of neurons, although hair-cell polarities are randomized. The lines representing terminals making ribbon synapses are graded in width to represent one or two synapses.

DOI: https://doi.org/10.7554/eLife.33988.010

The following source data and figure supplement are available for figure 5:

**Source data 1.** The tables provide a comprehensive set of measurements and calculations involving the data included in the main text and in *Figure 5—figure supplement 1*.

DOI: https://doi.org/10.7554/eLife.33988.012

**Figure supplement 1.** The diagrams depict the complete set of associations between hair cells and axonal terminals from three additional neuromasts (T2–T4) of *trilobite* mutants.

DOI: https://doi.org/10.7554/eLife.33988.011

We analyzed a transgenic line of zebrafish in which hair cells selectively express the Notch intracellular domain (NICD), the mediator of Notch signaling, and are thus regarded as Notch-overexpressing. All the hair cells in these larvae possessed anteriorly polarized hair bundles. We investigated by SBFSEM one mutant specimen from each of two animals (*Figure 6*; *Figure 6—figure supplement 1*; *Figure 6—source data 1*; *Video 4*). Grouping the hair cells by their anteroposterior positions after rearrangement revealed that 93% of presynaptic active zones were partnered with afferent terminals that were innervated by both anteriorly and posteriorly positioned hair cells. Furthermore, when hair cells were analyzed as sibling pairs, in every instance both cells innervated a common axonal terminal. In the absence of normal Notch signaling, the unique chemical identities mediating innervation apparently failed to be asymmetrically allotted to each sibling; instead, the same identity was possessed by both. Because we characterized the wiring and miswiring of neuromasts at 3–4 dpf, any subsequent alterations in wiring during the late larval, juvenile, or adult stages would have gone undetected.

Like the *trilobite* specimens, the Notch-overexpressing mutants also retained the wild-type microcircuit characteristics of redundancy and dominance even though the specificity of innervation had been eliminated. The transgenic neuromasts each showed only one highly dominant terminal as well as one of moderate dominance. Because the proportion of axonal terminals innervated by hair cells was the same as in normal neuromasts, high-dominance terminals with a preference for caudad-polarized cells were evidently able to displace some polarity-appropriate but low-dominance terminals. In the setting of hair cells uniformly possessing an anteriorly polarized phenotype, terminals of the opposing subset could thus outcompete the polarity-specific subset of fibers.

## Discussion

We have employed SBFSEM reconstructions to compare the development of the neuromast's microcircuit among wild-type individuals, mutants, and transgenic animals. The microcircuit is characterized by the progressive development of specificity, redundancy, and hierarchical dominance of innervation. Although previous research identified the phenomenon of polarity-specific afferent synapses in the neuromast, our approach details exquisite precision at the level of ribbon synapses as well as the degree of specificity among non-synaptic contacts and perisynaptic associations. We also find that the specificity of membrane contacts increases during hair-cell development; by analyzing *trilobite* mutants with random hair-bundle orientation, we establish that coincident hair-cell depolarization is neither necessary nor sufficient for specificity. Moreover, our data suggest that each daughter hair cell acquires a distinct biochemical identity that guides its innervation of axonal terminals.

In terms of redundancy we observe that the average hair cell forms ribbon synapses with two or more afferent terminals, develops membrane contacts with four afferent terminals, and is located near five or more afferent terminals within the perisynaptic compartment. We have also characterized efferent terminals, one of which occurs in each neuromast and contacts every mature hair cell. The fact that a single efferent axon innervates hair cells of both polarities suggests that activation of the efferent system

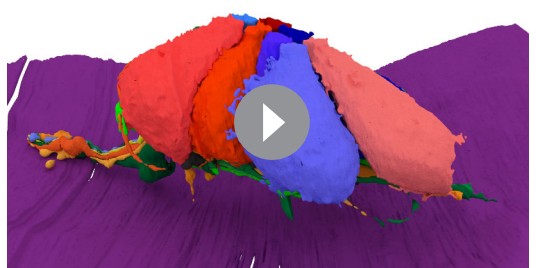

**Video 3.** Animated reconstruction of a *trilobite* mutant neuromast. This neuromast from a *trilobite* mutant specimen (*T1*) contains eight hair cells and eleven axonal terminals. The upper portions of the hair bundles were lost owing to fracture of the specimen block, but preimaging by confocal fluorescence microscopy sufficed for the assignment of bundle polarities. Hair cells are colored according to their anteroposterior positions following rearrangement: more anterior hair cells in reddish shades and more posterior hair cells in blue. The single efferent terminal is black. The basal lamina is colored purple. The spaces between the hair cells, and those around which the axonal terminals are arrayed, represent the positions of unseen supporting cells. In the second half of the sequence all the hair cells save one are removed to provide a view of the pattern of innervation, which is then further restricted to show the single dominant terminal. Occasional discontinuities in the terminals are an artifact of three-dimensional rendering rather than signifying an absence of data.

DOI: https://doi.org/10.7554/eLife.33988.013

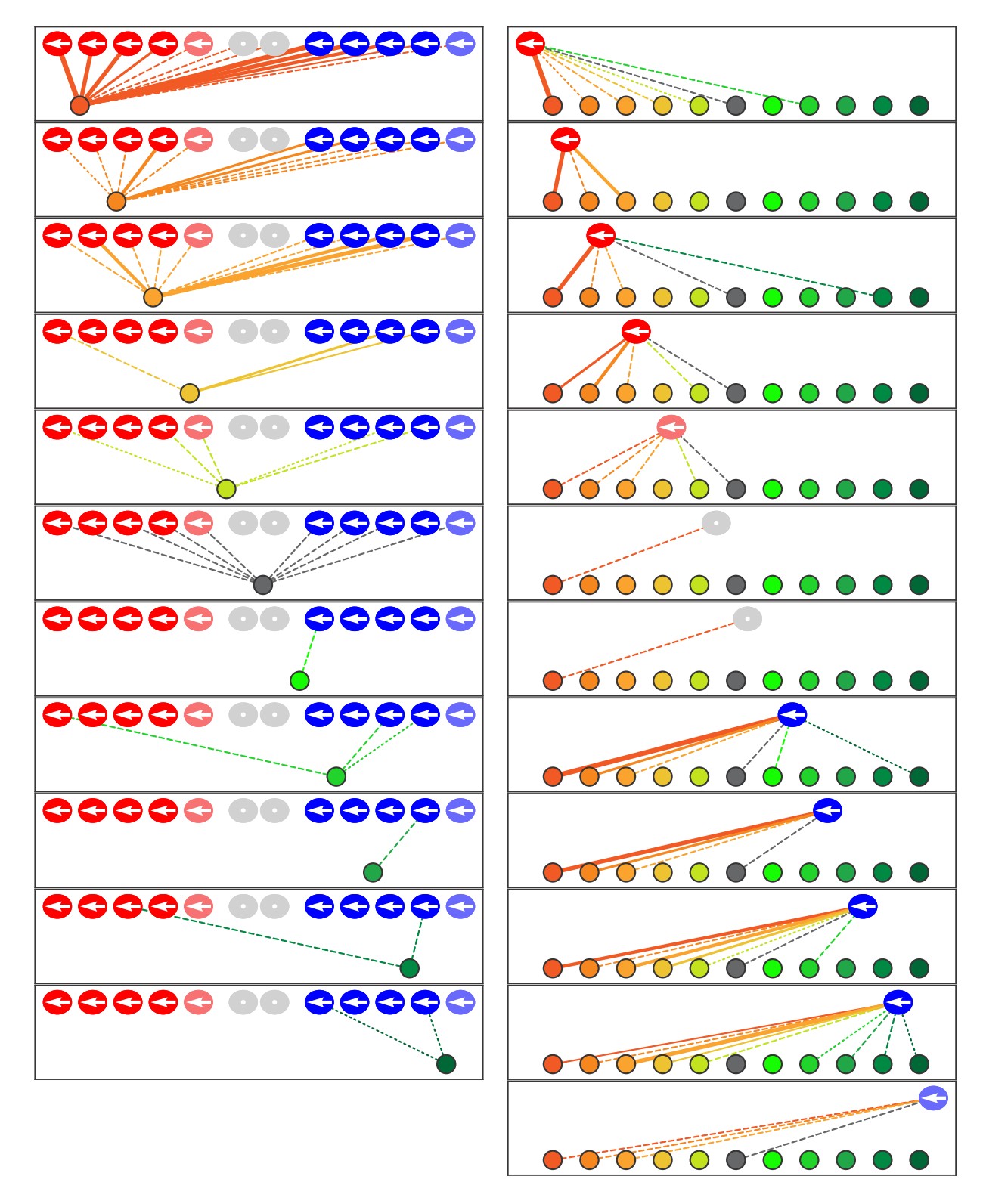

**Figure 6.** Microcircuit connectivity of the neuromast in Notch-overexpression mutant zebrafish. The diagram portrays the complete set of associations between axonal terminals and hair cells in a neuromast (N1) from a Notch-overexpression transgenic larva. Note the uniform polarization of the hair bundles for mechanosensitivity to rostrad stimuli. The arrangement and symbols are identical to those of *Figure 3*. Hair cells are divided into two groups according to their anteroposterior positions following rearrangement: more anterior hair cells colored in reddish shades while more posterior

*Figure 6 continued on next page*

*Figure 6 continued*

hair cells are blue. Because nascent hair cells 0–5 hr post-mitosis can be assigned neither a stable anteroposterior position nor a distinct hair-bundle orientation, they are marked as grey and located between the two hair-cell subpopulations. The lines representing the terminals making ribbon synapses are graded in width to represent from one to three synapses.

DOI: https://doi.org/10.7554/eLife.33988.014

The following source data and figure supplement are available for figure 6:

**Source data 1.** The tables provide a comprehensive set of measurements and calculations involving the data included in the main text and in *Figure 6—figure supplement 1*.

DOI: https://doi.org/10.7554/eLife.33988.016

**Figure supplement 1.** The diagram depicts the complete set of associations between hair cells and axonal terminals from one additional neuromast (N2) of a *Notch*-overexpression transgenic animal.

DOI: https://doi.org/10.7554/eLife.33988.015

nonspecifically desensitizes a neuromast, rather than adjusting its relative responsiveness to rostrad and caudad water movements. Such a system might, for example, suppress reflexes driven by the lateral line during an animal's own movements.

Despite the redundancy, the multiple terminals are distinctly unequal: across a majority of ribbon synapses and for most hair cells, there are varying levels of dominance of innervation and membrane contact. Among wild-type specimens, axonal terminals of a given polarity preference associate with ribbon synapses to the exclusion of more dominant terminals of the opposite preference. In Notch-overexpression larvae in which all hair cells adopt a common polarity, however, terminals of the opposite polarity preference participate in a limited number of ribbon synapses, suggesting that there are limits to the ability of polarity-specific afferent terminals to outcompete dominant ones of the opposite polarity.

By comparing the normal connectome to that of mutant zebrafish lines, we can identify systematic aberrations in the neuromast microcircuit arising from two cell-signaling pathways. The approach dissociates phenotypic characteristics that define hair-cell populations: the anteroposterior positioning of the somata relative to those of sibling cells, the angular orientation of the hair bundles, the anteroposterior orientation of the hair bundles, and distinct afferent-innervation partners. More specifically, we find that the canonical planar-cell-polarity pathway controls the angular orientation of hair bundles and affects their anterior-posterior positioning following rearrangement but does not affect the asymmetric allotment of the factor mediating polarity-specific innervation. In contrast, the Notch signaling pathway, while not necessary for angular orientation of the hair bundle, apparently mediates the asymmetric allotment of the factors responsible for anteroposterior hair-bundle polarization and innervation specificity. The rostrad polarity bias and wiring pattern of the Notch-overexpression mutants is remarkably similar to the phenotype observed in *Emx2* knockout fish (*Jiang et al., 2017*; *Ji et al., 2018*), providing further evidence that Emx2 is one of the asymmetrically alloted factors responsible for innervation specificity. To our knowledge, these results

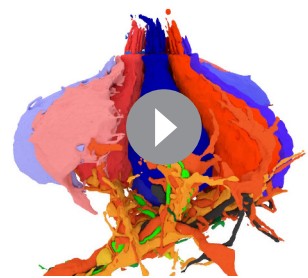

**Video 4.** Animated reconstruction of a neuromast from a Notch-overexpression transgenic animal. This neuromast from a Notch-overexpresion larva (*N1*) contains ten mature hair cells and one pair of immature hair cells along with eleven axonal terminals. Hair cells are colored according to their anteroposterior positions following rearrangement: more anterior hair cells in reddish shades and more posterior hair cells in blue. The terminals that form ribbon synapses with the former are delineated in various hues of orange and those associated with the latter are shaded green. The single efferent terminal is black. The spaces between the hair cells, and those around which the axonal terminals are arrayed, represent the positions of unseen supporting cells. In the second half of the sequence all the hair cells save one are removed to provide a view of the pattern of innervation, which is then further restricted to show the single dominant terminal. Occasional discontinuities in the terminals are an artifact of three-dimensional rendering rather than signifying an absence of data.

DOI: https://doi.org/10.7554/eLife.33988.017

constitute the first example of systematic neural miswiring delineated by SBFSEM.

Beyond the specific results reported in this study, the neuromast offers an excellent system for comparing connectomes across individuals, developmental stages, and genetic mutants. The analysis of neural circuits by SBFSEM has been hindered by the difficulty in identifying synapses, the absence of long-range information needed to identify specific types of axons and dendrites, and the enormous time required for both image acquisition and reconstruction. SBFSEM is facilitated in the neuromast, which offers well-delineated synapses and the ability to trace the entirety of each axonal terminal contained within the sensory unit. Each neuromast connectome represents 12 hr of microscopy and 500 hr of annotation, a fraction of the resources required by other efforts (*Helmstaedter et al., 2011*; *Wanner et al., 2016*; *Hildebrand et al., 2017*). Pairing recent advances in machine learning with our 528 billion annotated voxels of training data should further reduce the time required for reconstruction (*Berning et al., 2015*). Because the neuromast continues synaptogenesis throughout the larval period, it is possible to study synaptic development within a single data set. Finally, the analysis of synapse formation between two cellular subpopulations and approximately ten afferent neurons that is near perfect in its specificity offers a tractable vertebrate microcircuit for elucidating synaptic targeting in the nervous system. The nervous system of the zebrafish displays significant homology with that of humans, while offering many of the advantages of invertebrate model organisms: rapid development, transparent tissues, ease of genetic manipulations, and circumscribed neural connections.

The comprehensive examination of axonal terminals and their cellular environments within this self-contained sensory unit will be valuable for further characterizing the genes underlying neuronal competition and combinatorial innervation. The system may also be useful in defining the interactions between the canonical planar-cell-polarity pathway and Notch signaling as well as other pathways involved in hair-cell patterning. The intensive investigation of multiple neuromasts offers a useful complement to the delineation of the organs' long-range connectivity through large connectomic data sets (*Hildebrand et al., 2017*). With thousands of available transgenic zebrafish lines, including those targeting genes affecting hair-cell polarity and innervation (*Jiang et al., 2017*), we anticipate that comparative connectomics of the neuromast will offer valuable insights into the development of ribbon synapses in the inner ear and of vertebrate microcircuits in general.

## Materials and methods

### Nomenclature

The organization of neuromasts in the posterior lateral lines of zebrafish and other teleosts can cause confusion. Most of the hair cells at the anterior edge of each neuromast, and each of the mitotic daughter cells that adopts an anterior position, are morphologically polarized to be sensitive to water motion in the *posterior* direction. Conversely, the hair cells in relatively posterior positions are responsive to water movement in the *anterior* direction. The problem is confounded in some mutants, such as the *trilobite* line, in which the position of a hair cell within the neuromast and its direction of sensitivity are uncoupled.

To avoid confusion between cellular position and directional responsiveness, we use the term *rostrad*—toward an animal's rostrum, or front end—to denote water motions and mechanical sensitivity in that direction. The term *caudad* applies to water movements and sensitivity toward the caudum, or tail of an animal. In a wild-type larva, then, an anteriorly situated hair cell is usually sensitive to caudad water motion, whereas a posteriorly located hair cell generally responds to rostrad stimulation.

Several previous investigations have shown that each afferent axon is largely or completely sensitive to one direction of stimulation and that the sensitivity arises through selective innervation of that axon by hair cells of a common polarity. When discussing axonal terminals, we refer to their rostrad or caudad sensitivity on that basis. In other words, a terminal that makes several ribbon synapses with hair cells of a given morphological polarity is assumed to share the corresponding functional polarity.

## Animal care and breeding

Experiments were performed in accordance with the standards of Rockefeller University's Institutional Animal Care and Use Committee. Zebrafish were raised in E3 medium (5 mM NaCl, 0.17 mM KCl, 0.33 mM $CaCl_2$, 0.33 mM $MgSO_4$, 1 µg/mL methylene blue) in an incubator maintained at 28 ˚C. The experiments involved zebrafish of the *trilobite* mutant line and of the transgenic lines *Tg (myo6b:actin-GFP)*, *Tg(myo6:GFF)*, and *Tg(UAS:NICD-myc)* (*Hammerschmidt et al., 1996*; *Scheer and Campos-Ortega, 1999*; *Kindt et al., 2012*).

## Fluorescence imaging of zebrafish embryos

Throughout the lifespan of a zebrafish, mechanosensory hair cells arise continually within the neuromast to expand the total number of sensory cells and to replace senescent ones. In order to correlate hair-cell age with the SBFSEM data, we performed time-lapse confocal fluorescence microscopy for up to 39 hr on individual neuromasts prior to fixation.

The specific neuromasts examined, L1 to L5, were deposited by the first lateral-line primordium. Embryos of 1–3 dpf were anesthetized in 600 µM 3-aminobenzoic acid ethyl ester methanesulfonate in E3 medium and mounted in a 35 mm glass-bottomed dish in 1% low-melting-point agarose. Laser-scanning confocal imaging was performed with an inverted Zeiss Axio Observer Z1 with an LSM 780 system equipped with a 60X oil-immersion objective lens (Carl Zeiss, Oberkochen, Germany). Neuromasts were imaged at intervals of 45–60 min as Z-stacks acquired with 1.0 µm steps under laser excitation at 488 nm and 561 nm. Images were processed into hyperstacks and analyzed with FIJI (National Institutes of Health, Bethesda, MD). Each image depicted a plane of view tangential to the animal's surface and was oriented with anterior to the left and dorsal upward.

## Tissue preparation for SBFSEM

With the exceptions described below, SBFSEM was performed as reported previously (*Dow et al., 2015*). Following fluorescence imaging, the embryo was released from agarose at 2.5–4 dpf and placed within 30 min of the last image into 500 µM of fixative comprising 400 mM formaldehyde, 200 mM glutaraldehyde, 75 mM sodium cacodylate, 10 mM sucrose, and 1 mM $CaCl_2$. The specimen remained in fixative for 18 hr at 4°C before a single transverse cut was made approximately 200 µm posterior to the relevant neuromast.

All preparative steps remained as previously described except that the specimen was immersed in acetone following a graded series of ethanol washes and was embedded in Durcupan (Sigma-Aldrich, St. Louis, MN) following penetration of the specimen with graded mixtures of Durcupan and acetone that was enhanced by 40 s of microwave treatment at 150 W (Biowave, Pelco Corporation, Clovis, CA). The remainder of the protocol—curing the resin, mounting the specimen, and trimming the block face—followed the previous description (*Dow et al., 2015*). In total, we prepared eight neuromast specimens from eight wild-type zebrafish, four neuromasts from four *trilobite* zebrafish, and two neuromasts from two Notch-mutant zebrafish.

## SBFSEM and data analysis

Specimens were imaged at the Electron Microscopy Resource Center of Rockefeller University with a Merlin high-vacuum scanning electron microscope (Carl Zeiss, Oberkochen, Germany) equipped with a 3View2XP camera (Gatan Inc., Pleasanton, CA). The ultramicrotome shaved the block face in 30 nm increments and images were acquired with a lateral resolution of 6 nm X 6 nm per pixel. Alignment was performed with Digital Micrograph software (Gatan Inc.).

As described previously, data analysts were hired to contour cellular features within the SBFSEM data at their home computers. Gross features within the data sets, including hair cells and neurons, were identified by the lead scientist (ED) or project manager (SH) before being assigned for complete annotation by an analyst. The annotations were reviewed by the project manager for completeness and neatness. The results were additionally examined as a three-dimensional rendering to detect unlikely features in hair-cell somata or axonal terminals. Next, the annotations were reviewed in detail and edited by a senior data analyst or the project manager. Finally, all data were reviewed and edited by the lead scientist. Most annotation was performed accurately and precisely by the initial data analysts; editing generally involved the addition of small neurites or the resolution of complex or ambiguous features in the data.

## Quantification of neuromast features

The polarity of each hair cell, as defined by the orientation of its hair bundle, was determined by the position of the kinocilium relative to the stereocilia in the SBFSEM data and was confirmed by viewing the hair cell's apex in the confocal fluorescence-microscopic data.

Each presynaptic active zone was marked by a prominent synaptic ribbon. This spherical, electron-dense structure resided adjacent to the basolateral membrane of a hair cell and was surrounded by numerous synaptic vesicles. An axonal terminal was assigned as the synaptic partner of a ribbon if it was situated immediately across the synaptic gap from a presynaptic active zone. For instances in which the area opposite the ribbon was occupied equally by two terminals, each neuron was assigned half of the ribbon.

The contact areas between neurons and hair cells were determined automatically with a Python script developed with the computational-geometry package Shapely (*Toblerity.org, 2017*). We determined by visual inspection that our manual traces followed a given membrane with a precision of ±60 nm, a value determined by the nature of the manual annotation process and the thickness of membrane images. Our algorithm therefore considered two membranes to be in contact if their traces were separated by less than 60 nm and if no other trace intervened between them. We tested our contact algorithm against different samples throughout our data sets; the reported contact areas agreed consistently with the human assessments. An axonal terminal was scored as entering a neuromast if it branched from the posterior lateral-line nerve and crossed the basal lamina into the neuromast. Each such terminal was traced back to a single branch arising from a peripheral axon in the posterior lateral-line nerve.

## Classification of hair-cell age

Nine of the 14 specimens underwent timelapse confocal fluorescence microscopy for over 18 hr. We trained a machine-learning classifier on features of the hair cells in these data sets to assign age categories to the hair cells in the five data sets with only limited imaging. We obtained age information for 88 hair cells of a total of 149, randomly assigned 70% of these cells to a training set, and used the remaining 30% as a test set. Using these sets, we trained several classifiers using Scikit-Learn (*Pedregosa et al., 2011*), a Python package for machine learning. Each classifier was trained with the cellular-age category as an independent variable and the cell area, cell volume, total volume of ribbon synapses for each cell, and contact area of the neuromast's apex as dependent variables. Among the classifiers that we tested (Linear SVM, Gradient Boosting, Neural Network, Nearest Neighbors, Logistic Regression, Decision Tree, Naïve Bayes, and Random Forest), the Gradient Boosting and Decision Tree algorithms (*Hastie et al., 2009*) yielded the best classification scores against the test set. All assignments by the algorithm were consistent with phenotypic characteristics of the hair cells that are associated with developmental age.

## Statistical analysis

The significance of pairwise differences between groups of samples was computed by Student's two-tailed $t$-tests. For all figures, bars denote standard errors; $p < 0.05$ is indicated by *, $p < 0.01$ by **, and $p < 0.001$ by ***.

## Acknowledgements

We thank Ivana Mirkovic and Serhiy Pylawka for the *Tg(myo6b:GFF)* zebrafish line. Adedeji Afolalu provided outstanding zebrafish husbandry, Nadine Soplop conducted SBFSEM in the Electron Microscopy Resource Center, and the members of our research group offered useful comments on the manuscript. Agnik Dasgupta and Aaron Nagiel kindly contributed images for *Figure 1*. We made confocal-microscopic observations at the Bio-Imaging Resource Center. Extensive data annotation and quality assurance were performed by Kattya Grant and Zachary Tveter. The SBFSEM data were annotated by Angelina Almonte, Dennis Chan, Jennifer Cohen, Elizabeth Ephraim, Anna Eppel, Jocelyn Gan, Augustine Gnalian, Jeremy Griffith, Dinara Guliyeva, Daniel Hahn, Ediri Metitri, Alexandra Surman, and Anh Ung. The project was funded in part by a Project Grant from the Kavli Neural System Institute. ED was supported by the National Institute of Neurological Disorders and Stroke through a Ruth L Kirschstein National Research Service Award (DC013468) and by the National

Institute of General Medical Sciences through the Medical Scientist Training Program (GM07739). AJ was supported by the F M Kirby Foundation. KS was supported by the National Institute on Deafness and Other Communication Disorders through a Ruth L Kirschstein National Research Service Award (DC014212). AJH is an Investigator of Howard Hughes Medical Institute.

## Additional information

### Funding

| Funder | Grant reference number | Author |
|---|---|---|
| Howard Hughes Medical Institute | | A J Hudspeth |
| National Institute on Deafness and Other Communication Disorders | DC013468 | Eliot Dow |
| National Institute on Deafness and Other Communication Disorders | DC014212 | Kimberly Siletti |
| National Institute of General Medical Sciences | GM07739 | Eliot Dow |

The funders had no role in study design, data collection and interpretation, or the decision to submit the work for publication.

### Author contributions

Eliot Dow, Conceptualization, Supervision, Funding acquisition, Investigation, Methodology, Writing—original draft, Project administration, Writing—review and editing; Adrian Jacobo, Conceptualization, Data curation, Formal analysis, Supervision, Validation, Investigation, Visualization, Methodology, Writing—original draft, Writing—review and editing; Sajjad Hossain, Software, Formal analysis, Investigation, Methodology, Writing—original draft, Writing—review and editing; Kimberly Siletti, Supervision, Validation, Investigation, Visualization; A J Hudspeth, Resources, Methodology, Visualization, Writing - review and editing

### Author ORCIDs

Eliot Dow (iD) https://orcid.org/0000-0001-7409-2049
Adrian Jacobo (iD) https://orcid.org/0000-0001-9381-6292
A J Hudspeth (iD) https://orcid.org/0000-0002-0295-1323

### Ethics

Animal experimentation: Experiments were performed in accordance with the standards of Rockefeller University's Institutional Animal Care and Use Committee under protocol 16931.

### Decision letter and Author response

Decision letter https://doi.org/10.7554/eLife.33988.022
Author response https://doi.org/10.7554/eLife.33988.023

## Additional files

### Supplementary files

• Transparent reporting form
DOI: https://doi.org/10.7554/eLife.33988.018

### Data availability

The original data for 14 neuromasts are deposited at the webKnossos site https://demo.webknossos.org/, at which users can examine, manipulate, and download the data archived under the names

(e.g. "WT1") used in this publication. These records also constitute the source data for Figures 3, 5, and 6 and for Videos 1-4.

The following dataset was generated:

| Author(s) | Year | Dataset title | Dataset URL | Database, license, and accessibility information |
|---|---|---|---|---|
| Eliot Dow, Adrian Jacobo, Sajjad Hossain, Kimberly Siletti, A J Hudspeth | 2018 | Rockefeller_Neuromast_WT1 | https://demo.webknossos.org/datasets/Rockefeller_Neuromast_WT1/view#3546,4286,187,0,1.30 | Publicly available at Webknossos, to access, search for neuromast name e.g. WT1 |

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
