## [Decision Letter]

Thank you for submitting your article "Comparative connectomics of innervation in neuromasts of the zebrafish lateral line" for consideration by *eLife*. Your article has been reviewed by two peer reviewers, and the evaluation has been overseen by a Reviewing Editor and Eve Marder as the Senior Editor. The following individual involved in review of your submission has agreed to reveal his identity: Florian Engert (Reviewer #2).

The reviewers have discussed the reviews with one another and the Reviewing Editor has drafted this decision to help you prepare a revised submission.

As you will see from the reviews, both reviewers are in principle enthusiastic about the work. However, both reviewers raise similar issues regarding the presentation and analyses of the data, and clarity of the writing. Given the extent of the suggested revisions, both reviews are appended in full below. In particular, please ensure that the revision clearly states how the raw EM data and analyses files will be shared with the community.

Please remember that *eLife* has no limitations on the number of figures, and also uses figure supplements. So please take advantage of this to provide an excellent representation of your data.

Reviewer #1:

In this paper the authors use block face EM to analyse the microconnectivity of neuromasts in the zebrafish lateral line. The small volumes analysed and the throughput of the technique allows them to analyse microcircuits in several specimen, including mutant individuals. This is a very interesting study, but the authors need to work on the text and the presentation of the data.

Unfortunately, the manuscript places too much emphasis on the technique and describing the networks rather than addressing important biological questions. It is not a critique of the results, rather the way the authors wrote the paper. It is very clear for example from the closing sentence of the Abstract. It is not a conclusion of what was found, but a manifesto. Furthermore, 'comparative connectomics' is usually used to refer to inter-specific comparisons (see PMID: 23332749) and not to comparing the stereotypy of connections between individuals (which has already been done in *Drosophila* and *Platynereis* larvae).

It is not clear from the Introduction what motivated the study and how it helps us to interpret previous findings. Any reference to behavior and the relevance of the connectome to interpreting it are lacking from the discussion. For example, zebrafish larvae show rheotaxis (PMID: 28700578) that requires the integration of signals from anterior and posterior sensitive hair cells. The finding that nerve terminals are highly specific in innervating hair cells of one directional specificity is highly relevant to this behavior and should be discussed.

Unfortunately, the large amount of EM data are not presented in any detail. The authors should show volume renderings of reconstructed neuromasts to present for example how hair cells of one orientation connect to a specific subset of neuronal terminals.

The raw EM data should also be provided, at least for some specimen, as supplementary files (or deposited to the appropriate repositories, e.g. open connectome).

In general, more example EM images would be needed to show the different types of associations, the postsynaptic membrane cisternae and myriad presynaptic vesicles in efferent axon terminals, etc.

Likewise, the time-lapse fluorescence confocal microscopy data should be provided and also integrated into some of the figures, e.g. by providing the video, and showing snapshots in the figure, and then the corresponding EM reconstruction for the imaged neuromast, demonstrating that the cells can be assigned one-to-one between the datasets.

The mis-wiring results in the trilobite mutant and the Notch overexpressing fish are not discussed in any detail. The authors should come up with a summary diagram based on their results that places in cellular context Notch signaling, planar polarity, and synaptic wiring specificity.

The authors should include as Figure 1 a summary diagram of a fish with the lateral line neuromasts and indicate which ones were reconstructed, what is the size of the volumes, what is the general anatomy of a neuromast, how does it develop, etc. It is very hard to follow the paper otherwise for the non-specialists.

Reviewer #2:

The authors use serial block face electron microscopy to delineate the connectivity of afferent and efferent neurons within the zebrafish larva neuromast circuitry. They provide a comparative connectomic approach, both across neuromasts and genotypes. The data sets generated are impressive and clearly useful for the community. Overall, this is an important, novel and timely research project. However, there is a lot of room for improvement with respect to clarity and presentation.

The manuscript in its current form looks more like a sketch rather than a completed document:

1) There is no clear connection between the text and the figures. In particular, what is missing is a clear statement of each particular finding in the text, how this finding is corroborated by the data, and, with the help of specific and explicit figure panels, representative examples of such data need to be shown.

Furthermore it would be helpful to have cartoon-examples of main findings such as directional specificity, redundancy, dominance and hierarchy. These can then also be used to highlight the main differences in the two different genotypes.

2) An introduction to the neuromast system is missing. An overview figure with the known wiring diagram from the PLLG and the efferent nuclei are minimally required. This should also include a description of the known polarity specificity. Moreover, the relationship between the hair cells hair bundles position, flow sensitivity and age can clearly be illustrated. The classification of hair cells' mechano-sensitivity based on the position of hair-cells would also benefit from several examples drawn from the data.

3) The number of animals used and the number of neuromasts studied for each condition should clearly indicated.

4) The figures are not labelled and poorly referred to in the text. It is nearly impossible to figure out – in many cases – which figures are being talked about in the text. The connectivity diagrams in Figures 2, 5 and 6 are very hard to understand for a reader not familiar with neuromast connectivity. A graphic legend should be added to each figure explaining the different line sizes and line types. There are generally many ways in which the clarity of all figures can be improved. Every single one feels like a sketch with the essential data shown where little effort was made to guide the reader through the process of how the data was extracted and analysed. This makes the manuscript very difficult to read.

5) A 3D rendering of the reconstructions is missing, as well as illustrative "fly-throughs" across blocks of data – such videos are useful, essential and by now a standard way of displaying serial EM data.

6) The raw data, the segmented data and the primary analysis files all need to be shared with the community. It is not clear whether the authors plan to do so and, if yes, how.

7) A true strength of this study is the ability to quantify a neuron's "dominance" based on its occupation of presynaptic ribbons. The use dominance, however, needs to be clarified in the text because it refers to two types dominance (1) for a specific hair cell, and (2) for all hair cells of a single polarity. It is often confusing in the text to figure out which type of dominance is being discussed, and this is important in order to understand the authors' arguments on how this dominance arises (i.e. is it a local property at the level of a neuromast of or a property of the neuron as a whole?).

8) The authors use trilobite and Notch mutants to build clear arguments for the establishment of afferent specificity. However, since they highlight two other interesting metrics: redundancy and dominance, they should also discuss how these metrics are perturbed (or not) in the mutants in comparison to the wild-types. This comment pertains to the text overall: the utility of using "redundancy" and "dominance" to characterize the neuromast circuit should be demonstrated more clearly: What are the lessons we learn from these? Or, alternatively: What questions can we pose using these descriptors?

9) The Discussion needs to be a lot more specific in highlighting what new insights or findings, if any, were extracted from the data that go beyond what's already described in the existing literature.

---

## [Author Response]

Reviewer #1:In this paper the authors use block face EM to analyse the microconnectivity of neuromasts in the zebrafish lateral line. The small volumes analysed and the throughput of the technique allows them to analyse microcircuits in several specimen, including mutant individuals. This is a very interesting study, but the authors need to work on the text and the presentation of the data.Unfortunately, the manuscript places too much emphasis on the technique and describing the networks rather than addressing important biological questions. It is not a critique of the results, rather the way the authors wrote the paper. It is very clear for example from the closing sentence of the Abstract. It is not a conclusion of what was found, but a manifesto. Furthermore, 'comparative connectomics' is usually used to refer to inter-specific comparisons (see PMID: 23332749) and not to comparing the stereotypy of connections between individuals (which has already been done in Drosophila and Platynereis larvae).

The manuscript seeks to present a connectomic analysis of the microcircuit of the neuromast. To establish the basis for this novel approach, we believe that it is important to describe the technique and to delineate patterns of innervation in wild-type individuals that can be used for comparison to genetic mutants and subsequently to morphants and larvae exposed to different environment conditions or other manipulations. Having established this foundation, we apply the approach to important biological questions, namely to that of asymmetric cell division, its role in neuronal wiring, the decoupling of cellular identity and spatial position, and the relation of the canonical planar-cell-polarity pathway to other asymmetric-division pathways such as Notch. We recognize that the biological significance may not have received sufficient attention in some parts of the manuscript and we have sought to bring this to light.

The title has been changed so as not to confuse readers who might associate "comparative connectomics" with interspecific analyses.

It is not clear from the Introduction what motivated the study and how it helps us to interpret previous findings. Any reference to behavior and the relevance of the connectome to interpreting it are lacking from the discussion. For example, zebrafish larvae show rheotaxis (PMID: 28700578) that requires the integration of signals from anterior and posterior sensitive hair cells. The finding that nerve terminals are highly specific in innervating hair cells of one directional specificity is highly relevant to this behavior and should be discussed.

We have added this information to the Introduction.

Unfortunately, the large amount of EM data are not presented in any detail. The authors should show volume renderings of reconstructed neuromasts to present for example how hair cells of one orientation connect to a specific subset of neuronal terminals.

We have added Figure 2 to provide examples of SBFSEM images of a representative neuromast and of the specific features scored in our analysis of the interactions between hair cells and nerve terminals. Moreover, we have included three dynamic videos depicting volume renderings of representative data.

The raw EM data should also be provided, at least for some specimen, as supplementary files (or deposited to the appropriate repositories, e.g. open connectome).In general, more example EM images would be needed to show the different types of associations, the postsynaptic membrane cisternae and myriad presynaptic vesicles in efferent axon terminals, etc.

As noted above, we have added Figure 2 to provide examples of SBFSEM images of a representative neuromast and of the specific features scored in our analysis of the interactions between hair cells and nerve terminals. When the manuscript has been accepted, we shall make 18 complete neuromast data sets and annotations publicly available to navigate or download at the webKnossos site hosted by scalableminds.

Likewise, the time-lapse fluorescence confocal microscopy data should be provided and also integrated into some of the figures, e.g. by providing the video, and showing snapshots in the figure, and then the corresponding EM reconstruction for the imaged neuromast, demonstrating that the cells can be assigned one-to-one between the datasets.

We have created an additional video that features images from time-lapse confocal fluorescence microscopy and the correlation of hair-cell identity with SBFSEM data.

The mis-wiring results in the trilobite mutant and the Notch overexpressing fish are not discussed in any detail. The authors should come up with a summary diagram based on their results that places in cellular context Notch signaling, planar polarity, and synaptic wiring specificity.

Although we would like to offer a definitive response to this suggestion, there are several questions remaining to be answered that preclude a summary of the mediation of cellular innervation, planar-cell polarization, orientation of the hair bundle, and cellular rearrangement by signaling pathways. We are currently preparing another manuscript that incorporates many additional experiments bearing on these issues.

The authors should include as Figure 1 a summary diagram of a fish with the lateral line neuromasts and indicate which ones were reconstructed, what is the size of the volumes, what is the general anatomy of a neuromast, how does it develop, etc. It is very hard to follow the paper otherwise for the non-specialists.

We have accepted this useful suggestion and have prepared a Figure 1 with the requested information.

Reviewer #2:The authors use serial block face electron microscopy to delineate the connectivity of afferent and efferent neurons within the zebrafish larva neuromast circuitry. They provide a comparative connectomic approach, both across neuromasts and genotypes. The data sets generated are impressive and clearly useful for the community. Overall, this is an important, novel and timely research project. However, there is a lot of room for improvement with respect to clarity and presentation.The manuscript in its current form looks more like a sketch rather than a completed document:1) There is no clear connection between the text and the figures. In particular, what is missing is a clear statement of each particular finding in the text, how this finding is corroborated by the data, and, with the help of specific and explicit figure panels, representative examples of such data need to be shown.Furthermore it would be helpful to have cartoon-examples of main findings such as directional specificity, redundancy, dominance and hierarchy. These can then also be used to highlight the main differences in the two different genotypes.

There were submission errors that resulted in figures being out of order and otherwise incorrectly referenced in the text. We apologize to the reviewers for these mistakes. After correcting these errors, we have extensively revised the figures and text to create a closer connection between the two. We have included figures that directly display characteristics of the microcircuit as well as a descriptive figure illustrating the major features of the lateral-line neuromast.

2) An introduction to the neuromast system is missing. An overview figure with the known wiring diagram from the PLLG and the efferent nuclei are minimally required. This should also include a description of the known polarity specificity. Moreover, the relationship between the hair cells hair bundles position, flow sensitivity and age can clearly be illustrated. The classification of hair cells' mechano-sensitivity based on the position of hair-cells would also benefit from several examples drawn from the data.

We have added Figure 1, which includes most of the points specified by the reviewer with an emphasis on hair-cell polarity and polarity-specific innervation, the major characteristics explored in our investigation.

3) The number of animals used and the number of neuromasts studied for each condition should clearly indicated.

We have revised the manuscript to make clear the number of specimens and individuals both in the Results and the Materials and methods.

4) The figures are not labelled and poorly referred to in the text. It is nearly impossible to figure out – in many cases – which figures are being talked about in the text. The connectivity diagrams in Figures 2, 5 and 6 are very hard to understand for a reader not familiar with neuromast connectivity. A graphic legend should be added to each figure explaining the different line sizes and line types. There are generally many ways in which the clarity of all figures can be improved. Every single one feels like a sketch with the essential data shown where little effort was made to guide the reader through the process of how the data was extracted and analysed. This makes the manuscript very difficult to read.

We have added Figure 2 to show the general structure of a neuromast and its components and to provide examples of the various types of interaction between a hair cell and the associated afferent nerve terminals. At the bottom left of Figure 3 we have included a key to the cellular positions, hair-bundle orientations, afferent and efferent neuronal identifies, and types of interaction between hair cells and afferent axonal terminals.

5) A 3D rendering of the reconstructions is missing, as well as illustrative "fly-throughs" across blocks of data – such videos are useful, essential and by now a standard way of displaying serial EM data.

We have created dynamic videos depicting volume renderings of representative data for a wild-type larva, a *trilobite* mutant, and a Notch-overexpressing transgenic larva.

6) The raw data, the segmented data and the primary analysis files all need to be shared with the community. It is not clear whether the authors plan to do so and, if yes, how.

We fully intend to make the neuromast data sets and annotations publicly available for examination and downloading. Unfortunately, there is as yet no standard means of sharing such data and no publicly available repository for doing so. The best solution we have found is to set up a repository using the platform developed by scalableminds, which specializes in hosting SBFSEM data sets. We have retained their services and are working with them to make the data available; we shall provide a link when the manuscript has been accepted.

7) A true strength of this study is the ability to quantify a neuron's "dominance" based on its occupation of presynaptic ribbons. The use dominance, however, needs to be clarified in the text because it refers to two types dominance (1) for a specific hair cell, and (2) for all hair cells of a single polarity. It is often confusing in the text to figure out which type of dominance is being discussed, and this is important in order to understand the authors' arguments on how this dominance arises (i.e. is it a local property at the level of a neuromast of or a property of the neuron as a whole?).

The question of whether a neuron possesses dominance across neuromasts or only within a neuromast is interesting and deserves further study. Because only individual neuromasts were interrogated in this study, however, we are unable to drawn such conclusions. We have mentioned anecdotal data from published studies of single-fiber labeling that may relate to the matter. In revising the manuscript we have sought to clarify the issue of neuronal dominance for an individual hair cell versus dominance across multiple hair cells.

8) The authors use trilobite and Notch mutants to build clear arguments for the establishment of afferent specificity. However, since they highlight two other interesting metrics: redundancy and dominance, they should also discuss how these metrics are perturbed (or not) in the mutants in comparison to the wild-types.

As mentioned in the text, SBFSEM has allowed us to conclude that although specificity is affected in distinct ways in the *trilobite* mutants and Notch transgenics, the redundancy and dominance are not.

This comment pertains to the text overall: the utility of using "redundancy" and "dominance" to characterize the neuromast circuit should be demonstrated more clearly: What are the lessons we learn from these? Or, alternatively: What questions can we pose using these descriptors?

We have amended the language throughout the manuscript to better highlight the meaning and utility of “redundancy," the multiple terminals associating at each level with a hair cell, and “dominance," the hierarchical nature of the multiple associations. One could pose many questions focusing on these terms. Would a terminal exert greater dominance at a neighboring or new hair cell if its associations at another hair cell were eliminated? Might that terminal's status in the dominance hierarchy of one neuromast increase if its associations at a second neuromast were eliminated? Does a redundantly innervated terminal assume the synaptic associations of another terminal that has been destroyed, or does a third terminal supervene? Could redundant innervation lead to the differential response of fibers to mechanical deflection of the hair bundle for more finely graded characterization of stimuli? What are the molecular pathways that mediate the competition of terminals in the neuromast, and by over-activating or eliminating a factor can we render a terminal more or less dominant? We hope that our initial publication will provide a set of baseline observations against which the results of these procedures can be tested.

9) The Discussion needs to be a lot more specific in highlighting what new insights or findings, if any, were extracted from the data that go beyond what's already described in the existing literature.

We have expanded the Discussion to highlight the main findings of the research.